# The Effect of the Addition of Low-Alkaloid Lupine Flour on the Glycemic Index In Vivo and the Physicochemical Properties and Cooking Quality of Durum Wheat Pasta

**DOI:** 10.3390/foods11203216

**Published:** 2022-10-14

**Authors:** Ada Krawęcka, Aldona Sobota, Piotr Zarzycki

**Affiliations:** Division of Engineering and Cereals Technology, Department of Plant Food Technology and Gastronomy, University of Life Sciences in Lublin, Skromna 8 Street, 20-704 Lublin, Poland

**Keywords:** pasta, functional food, low GI, low GL

## Abstract

The aim of this research was to determine the effect of the addition of lupine flour (LF) on the values of the glycemic index (GI) and glycemic load (GL), physicochemical properties, and cooking quality of durum semolina pasta. The pasta was enriched with 0–25% of lupine flour (LF0-LF25). Additionally, 7.5 and 20% of oat β-glucans, 5% of vital gluten, and 20% of millet flour were used in a selected sample. The addition of 7.5% β-glucans and 5% vital gluten to the product resulted in only a slight decrease in the GI of the products. A significant decrease in pasta GI was noted after the addition of 20% of lupine flour. The product enriched with 20% of lupine flour, 20% of β-glucans, and 20% of millet flour had the lowest glycemic index and glycemic load (GI = 33.75%, GL = 7.2%, respectively). At the same time, the lupine-flour-enriched products were characterized by an increased concentration of protein, fat, ash, and dietary fiber. The addition of lupine flour at the level of up to 20% yielded functional products characterized by good cooking quality.

## 1. Introduction

A diet with a low glycemic index (GI) and glycemic load (GL) is mainly advised for people with diabetes, but is also crucial for people with metabolic diseases to help prevent the development of type 2 diabetes. Scientific research shows that it also helps to reduce the risk of cardiovascular diseases and some cancers and is an important support in the treatment and prevention of obesity [1]. Glycemic index (GI) describes the blood glucose response after the consumption of a carbohydrate-containing test food relative to a carbohydrate-containing reference food (glucose or white bread). The concept of glycemic load (GL) considers the GI of a food and the amount eaten [2,3].

Pasta is a widely consumed and popular grain product. It has a long shelf-life and is easy to prepare. Our previous research shows that extruded pasta produced on the basis of durum semolina is characterized by a low glycemic index [4]. A meta-analysis by Chiavaroli et al. [5] indicates that pasta can be successfully used in the context of low-GI dietary patterns. The compact and dense structure of products, in which the gluten network surrounds starch granules, is primarily responsible for the low GI of pasta. This reduces the enzymatic susceptibility of starch and contributes to its slow digestion [2,6]. Ingredients classified as soluble dietary fiber, including β-glucans from oats and barley, have a documented and confirmed hypoglycemic effect [7,8]. It should be noted that the substitution of semolina with unconventional raw materials containing dietary fiber can weaken the protein–starch matrix and have a negative effect on the cooking and sensory qualities of pasta [9]. The negative impact of high-fiber additives can be reduced by introducing vital gluten into the product. Research conducted by Krawęcka, Sobota, and Sykut-Domańska [10] has proved that the use of 5% vital gluten addition strengthens the structure of the gluten network in fiber-enriched pasta and yields a high-quality product.

Research indicates that white lupine seeds (*Lupinus albus,* L.) may be an interesting raw material with a positive effect on postprandial glycemia [11,12]. For years, attention has been paid to the desirable nutritional properties of the seeds of this legume, as well as the ease and low cost of growing the plant. White lupine seeds are primarily rich in protein (40%) and fiber (up to 39%). They contain about 8–12% of fat rich in unsaturated fatty acids, mainly α-linolenic acid and sitosterols. Compared to other legumes, lupine seeds contain small amounts of digestible carbohydrates (3.5%) and are a rich source of minerals and biologically active substances, e.g., tocopherols, carotenoids, phenolic acids, and flavonoids [13,14]. The protein present in lupine has a high biological value, comparable to that of eggs. It is characterized by a high essential amino acid index (EAAI) and protein efficiency ratio (PER), based on lysine and tyrosine availability [15]. Sipsas [16] emphasizes that lupine proteins have excellent apparent digestability (80.0–85.8%), similar to that of casein protein (87.1%). The addition of lupine to wheat flour contributes to an increase in not only the protein concentration in the product but also its biological value. In terms of the content of essential amino acids, wheat and lupine proteins seem to be complementary. Lupines are rich in essential amino acids, particularly phenylalanine, tyrosine, and lysine, but deficient in sulfur amino acids (methionine and cystine), while wheat protein is characterized by the opposite dependence [13]. In lupine seeds, globulins represented by α-, β-, γ-, δ-conglutins are the dominant protein fraction (85–88% of protein). Compared to globulins, the content of albumin is nine-fold lower, and prolamines are only present in small amounts. As reported by Duranti et al. [12], lupine globulins have a multidirectional beneficial effect on the human body. The authors emphasize that γ-conglutins have the ability to lower blood glucose and insulin levels, and the other fractions of globulins lower the level of cholesterol and triglycerides in the blood and regulate blood pressure [17]. It should be noted that lupine seeds also contain anti-nutritional substances, including oligosaccharides and alkaloids responsible for the characteristic bitter taste. The low-alkaloid lupine varieties, referred to as sweet, are used for food purposes. The content of alkaloids in these varieties should not exceed 500 mg/kg of seeds [18,19]. In some countries (Australia, New Zealand, France, UK), the limit is 200 mg alkaloids/kg of lupine flour or products containing lupine seeds [20]. An effective strategy, which might remove the a-galactosides from the lupin flour, is to isolate only the protein fraction to produce a lupin protein isolate. In addition to lupine flour, the substitution of semolina with millet may also positively influence the GI value in pasta products. Millet is known for its low starch digestibility, which results from, e.g., the simple methods of processing the raw material and the presence of fiber, polyphenols, and unsaturated fatty acids [21]. Millet flour has a semolina-like appearance and is delicate in flavor. It has been suspected that it may neutralize the flavor of the pulses. Shukla and Srivastava [22] assesed the impact of finger millet on the nutritive value and GI of pasta. Authors noted a significant reduction in pasta GI after replacing 30% of wheat flour with millet (45.13% for fortified pasta vs. 62.59 for control). Millet flour seems to be a valuable substitute for semolina in the context of its influence on glycaemia, so it was part of the research scope.

There are reports in the literature on the enrichment of cereal products with lupine flour. Attempts were made to enrich wheat flour pasta with lupine flour, and pasta with an increased protein content was obtained [23]. To date, gluten-free pasta has also been successfully produced, based on rice flour with the addition of eggs, lupine flour, and guar gum, with improved nutritional value [24]. Lupine was also used to enrich wheat bread [25,26]. However, there are studies determining the effect of the addition of lupine protein isolate on in vitro starch digestion [27]. It seems advisable to develop functional pasta with low GI and GL values confirmed by in vivo tests, which could be useful for people with disturbed carbohydrate metabolism. The aim of the study was to determine the possibility of using lupine flour and, additionally, oat β-glucans, vital gluten, and millet flour to obtain functional pasta with a low GI and GL and high cooking quality. The literature data indicate that the above-mentioned raw materials can reduce the GI of products.

## 2. Material and Methods

### 2.1. Characteristics of Raw Materials

Durum semolina (Julia Malom, Kunszállás, Hungary) was used as the main ingredient for the production of pasta. The other ingredients were lupine flour (VegaProvita, Sviadnov, Czech Republic), in which the content of alkaloids (quinolizidine and indole) did not exceed the permissible standard of 0.2 g/kg dm, according to the certificate, vital gluten (Młyn Niedźwiady, Poland), oat β-glucans (Brenntag, Kędzierzyn-Koźle, Poland), and millet flour (Młyn Niedźwiady, Niedźwiady, Poland). The control sample (CON) was made of durum semolina. The addition of lupine flour varied, and amounted to 0, 5, 10, 15, 20, and 25% (samples LF0, LF5, LF10, LF15, LF20, and LF25, respectively). For the LF0-LF25 trials, a constant dose of β-glucans (7.5%) and vital gluten (5%) was provided. In the LFM20 sample, 20% of millet flour was used, in addition to lupine flour (20%), oat β-glucans (20%), and vital gluten (5%). A detailed model of the experiment is presented in Table 1.

### 2.2. Pasta Preparation

A MAC-30S Lab pasta extruder (ItalPast, Fidenza, Italy) and an EAC30-LAB pasta dryer (ItalPast, Fidenza, Italy) were used for the production of pasta in semi-technical conditions. The ingredients provided by the recipe were dosed into the pasta extruder mixer and mixed for 15 min. After this time, the dough was transferred to the bottom mixer tank and mixing was continued under vacuum. At the same time, the extruder screw was started, and fusilli-shaped pasta was extruded. The rotational speed of the screw of the pasta extruder was 48 rpm. A Teflon die was used. The pasta samples were dried at a controlled temperature and relative air humidity. Detailed conditions of this process were described previously by Sobota et al. [28].

### 2.3. Chemical Analysis

The chemical composition of the raw materials and pasta samples was performed using the AACC and AOAC methods [29,30].

Moisture content and ash content was determined using the AACC method 44-15A and 08-01, respectively [29].

Protein content was measured using the Kjeldahl method using the Kjeltec 2300 analyzer (FOSS, Höganӓs, Sweden) (Method AACC 46-08). The protein content was calculated from the total nitrogen content with the use of a conversion factor of 5.7.

Fat content was determined by continuous extraction in SoxtecTM8000 (FOSS, Höganäs, Sweden) with hexane as a solvent.

Total dietary fiber (TDF), insoluble dietary fiber (IDF), and soluble dietary fiber (SDF) contents were determined according to the enzymatic methods (AACC 32-05, AACC 32-21, AOAC 991.43, and AOAC 985.29) as described Krawęcka et al. [10].

Digestible carbohydrate content was calculated from the difference by subtracting the sum of all macronutrients (protein, fat, ash, total dietary fibre) from 100, according to Krawęcka et al. [4].

### 2.4. Physical Properties

**Apparent viscosity** was tested using rotary rheometer RM 180 (Mettler-Toledo AG, Switzerland, software RSI Orchestrator, ver. V6.5.8.), in accordance with the method developed by Zarzycki and Sobota [31].

### 2.5. Cooking Quality of Pasta Samples

**Optimum cooking time (OCT)** and c**ooking loss** (**CL**, g/100 g d.m.) were measured according to Method AACC 66-50.01 [29].

The weight increase index (WII) and the volume increase index (VII) was determined, as described by Krawęcka et al. [4].

### 2.6. Determination of Glycemic Index (GI) In Vivo and Glycemic Load (GL)

GI and GL were evaluated for selected pasta samples. To determine the GI of pasta and the effect of consumption of this product on the postprandial glucose value, measurements of capillary blood glucose were performed in 12 volunteers. The basis for the research was the consent from the Committee on Ethics of Human Scientific Research at the Faculty of Human Nutrition and Consumer Sciences of Warsaw University of Life Sciences (WULS-SGGW) (Resolution No. 18/2018). Volunteers qualified for the test after prior medical consultation and after offering written consent to participate in the study. The participants were informed about the purpose and scope of the study and properly instructed to maintain a proper diet the day before the study. The glucose concentration in capillary whole-blood was determined with the dry enzymatic method using test strips and the Roche Accutrend GCT apparatus. Capillary blood was collected with disposable lancets after disinfecting the fingertip. Each time, the participants of the study received an aqueous solution of 50 g of glucose or an amount of pasta providing 50 g of digestible carbohydrates and consumed within 10 min. Subsequent glucose concentration measurements were taken from the end of consumption every 15 min in the first hour and every 30 min in the next hour. The measurements were carried out in duplicate:-After consuming the solution containing 50 g of glucose as a standard;-After consuming an amount of traditional pasta providing 50 g of digestible carbohydrates;-After consuming an amount of experimental pasta providing 50 g of digestible carbohydrates.

Two extreme measurement results for the highest and lowest postprandial glycemia were rejected, and the results from 10 participants were used for further analysis. To calculate the glycemia of a given product, a glycemic curve was plotted, connecting the points indicating the concentration of glucose in the fasting capillary blood and determined after consumption of the tested product every 15 min in the first hour and every 30 min in the next hour. Fasting glucose was taken as the baseline level. By summing up the areas of individual trapezoids and triangles, the area under the glucose curve was calculated [32].

The glycemic index values were calculated by relating the blood glucose value of the tested product calculated from the area under the curve (IUAC of the tested products) to the area under the glycemic curve for the standard (glucose) (IUAC) and multiplying the result by 100%.
Glycemic index (%) = (IAUC of tested product/IAUC of glucose standard) × 100%

Glycemic load was calculated from the formula:GL = (W * GI)/100
where: W—the amount of carbohydrates in a standard portion of the product (50 g) [g]; GI—glycemic index determined in the study (%).

### 2.7. Statistical Analysis

For the statistical analysis, the STATISTICA 13.1 (StatSoft ©, Inc., Tulsa, OK, USA) software was used. All experimental results were means (± S.D) from at least three assays. One-way analysis of variance (ANOVA) and Tukey’s posthoc test were used to compare the groups. The results were statistically different for *p*-values ≤ 0.05.

## 3. Results and Discussion

### 3.1. Pasta Processing

The addition of lupine flour, as well as oat β-glucans, vital gluten, and millet flour, affected the pasta extrusion process. The use of oat β-glucans and vital gluten in the trials (LF0, LF5, LF10, LF15, LF20, LF25, LFM20) resulted in higher-pressure values compared to the control sample (CON) (Table 1). The pressure increased from 8.5 MPa for the CON sample to 13.5 MPa for the other samples. The introduction of high-fiber ingredients and gluten makes the dough harder and less plastic, and limits the dough flow rate through the formation of holes in the die. The addition of gluten, which induces the formation of a stronger starch-protein matrix of pasta, further increases this trend. Similar results were obtained by Krawęcka, Sobota, and Sykut-Domańska [10]. The authors found that the substitution of semolina with 5% of vital wheat gluten and 5% of xanthan gum was accompanied by an increase in the discharge pressure from 8.5 to 13 Mpa [10]. In turn, Petitot et al. [33] observed a reduction in pressure when pressing noodles enriched with chickpea flour. Zarzycki et al. [34], who added high-fiber raw materials (flaxseed flour and cake) into pasta, noted a drop in pressing pressure. It should be assumed that the fat content in flaxseed raw material, which ensures a greater plasticity of dough, may be of great importance in shaping the consistency of dough and the pressure in the pasta pressing process.

### 3.2. Chemical Analysis

The chemical composition of the pasta samples is shown in Table 2. The addition of lupine flour caused a significant (*p* ≤ 0.05) increase in the protein content. In the sample enriched with the 25% addition of lupine flour (LF25), the protein content was 68% higher than in the control sample (CON). In turn, an increase in the share of oat β-glucans from 7.5 to 20% (LF20 and LFM20 samples, respectively), characterized by a relatively low protein content (8.45%), resulted in a lower protein concentration in the pasta. A similar relationship was observed in a study on the effect of the addition of oat β-glucans on the physicochemical properties of pasta [10], where an increase in the share of β-glucans (up to 20%) was accompanied by a significant (*p* ≤ 0.05) decrease in the protein content. One of the most important elements that influence the quality of pasta is the gluten network surrounding the starch granules. A higher-gluten protein content promotes the formation of a strong protein matrix [35], but the introduction of legume proteins, which are mainly in fractions of albumin and globulin [36], may be associated with the formation of a weaker gluten matrix. With this in mind, protein-rich vital gluten was introduced into products enriched with legumes, which supported the formation of a stable gluten network.

Martínez-Villaluenga et al. [37] attempted to obtain pasta from semolina with the addition of alcohol-extracted lupine flour, as well as pasta supplemented with fermented and sprouted pigeon peas. They obtained products with a significantly improved quality of protein. The authors confirm that lupine proteins can be a good supplement for semolina proteins due to the higher content of lysine and lower levels of methionine. However, the use of alcohol extraction resulted in a reduction in the content of γ-conglutin in lupine flour. It should be emphasized that this protein fraction is attributed with a hypoglycemic effect; hence, it is advisable to use raw lupine flour to develop recipes for products with low GI.

In the present study, along with the increase in the lupine flour amount, the ash and fat content significantly increased (*p* ≤ 0.05), proportionally with the share of the legume seeds. There were no significant differences between the LFM20 sample with the addition of millet flour and the LF15-LF25 sample in terms of the ash content. However, the LFM20 sample significantly differed (*p* ≤ 0.05) from all the other variants, as it had the highest fat content. Increased ash content may indicate a greater amount of minerals in the products, which may be accompanied by a greater amount of fiber [4]. Fat can have a positive effect on the cooking quality of pasta, increasing the stability of the starch–protein matrix [38].

A significant (*p* ≤ 0.05) increase in the content of total fiber (TDF) was noted along with the increase in the share of lupine flour. Moreover, the large share of the high-fiber raw material) in the LFM20 sample significantly reinforced this trend (*p* ≤ 0.05). The values of TDF in the LF25 and LFM20 samples were over four- and fivefold higher than in the control sample (CON), respectively. The value of SDF was significantly (*p* ≤ 0.05) higher in the sample supplemented with millet flour (LFM20) than in the other samples containing lupine flour and the control sample (CON). This was related to the highest proportion of oat β-glucans, which contain as much as 32.55% d.m. SDF. Significant (*p* ≤ 0.05) differences in the IDF content were noted for samples LF20 and LF25, whose values were almost five-fold higher than in the control sample (CON). The increased content of dietary fiber in the samples enriched with lupine flour resulted in a significant (*p* ≤ 0.05) reduction in the content of available carbohydrates compared to the control sample (CON). It should be emphasized that, under pasta cooking scenarios, dietary fiber competes with starch granules for water availability and reduces the swelling and gelation of starch, which may decrease its digestibility [39].

### 3.3. Physical Properties

The addition of lupine flour did not significantly affect the viscosity of the product (Table 3). The main determinant of pasta viscosity was the share of β-glucans. The LFM20 sample, containing the largest 20% addition of oat β-glucans, was characterized by the highest viscosity during heating to 95 °C and after cooling. No significant differences in viscosity were noted in the other variants at both the heating and cooling stage for the pasta gruels. Similar results were obtained by Krawęcka et al. [10], who enriched semolina pasta with oat β-glucans and xanthan gum. The authors proved that an increase in the proportion of components rich in a soluble fiber fraction causes an increase in viscosity at both the heating stage and after cooling the gruel. Increased viscosity may be of great importance for the functional properties of enriched pasta. Increased viscosity in the intestine delays glucose and cholesterol absorption and inhibits bile acid reabsorption [40].

### 3.4. Cooking Quality

The research showed that samples enriched with the 20–25% addition of lupine flour (LF20, LF25, and LFM20), compared to the other pasta samples, require a much longer optimum cooking time (OCT) (11.5–12 min) (Table 4). The OCT of products enriched with β-glucans and vital gluten (LF0) alone and lupine flour at the level of 5–15% (samples LF5-LF15) was 9–10 min, and did not significantly differ (*p* ≤ 0.05). Similar results were obtained by Krawęcka et al. [10], who enriched pasta made of durum semolina with oat β-glucans and xanthan gum. In this case, the 15–20% addition of β-glucans caused a significant elongation of OCT, while there were no significant differences in OCT between the control and the samples enriched with 5–10% of oat β-glucans. The observed OCT changes may be caused by the high water absorption by high-fiber components in products, which compete for water with starch, thereby hindering its swelling and pasting [9,34]. A reverse tendency was noted by Jayasena and Nasar-Abbas [41]. The authors observed a significant decrease in the OCT of pasta with a 40% replacement of semolina with lupine flour. This large addition of legumes could significantly weaken the gluten network and, consequently, facilitate starch gelatinization. With a 10% and 20% share of lupine flour, they did not notice any significant differences in relation to the control sample.

The highest CL was recorded for the LFM20 and LF25 samples with the highest addition of non-gluten flours (lupine and millet flours). The higher CL values may be related to the weakened gluten matrix in the enriched pasta. A similar tendency was noted by Petitot et al. [33] and Teterycz et al. [42], who enriched pasta made of durum semolina with flour from legume seeds. Progressive CL were also visible with an increase in the share of millet flour in semolina pasta [43]. However, the loss of dry matter in all the pasta samples with the addition of lupine flour was relatively low and did not exceed 8% of d.m. Similar low CL values were obtained by Cutillo et al. [44] for spaghetti enriched with a lupine protein isolate up to 30%. The authors concluded that the extrusion method has a great influence on the CL. A higher CL (up to 22%) was noted for spaghetti produced by twin-screw extrusion than by single-screw extrusion. In our study, even in the case of pasta enriched with 25% lupine flour, CL did not exceed 8%; however, the appearance of the cooked product (Figure 1) proves that this addition is too high. Pasta LF25 does not retain its shape after cooking and becomes deformed. Previous studies have shown that a high-quality product can be obtained by substituting semolina with lupine flour at levels of up to 10% [44].

The addition of lupine flour up to 25% did not produce an increase in the pasta volume index. The differences in the weight increase index in most of the studied samples were statistically insignificant.

### 3.5. Determination of Glycemic Index (GI) In Vivo and Glycemic Load (GL)

Changes in blood glucose levels after the consumption of a standard amount of glucose (50 g) and an appropriate portion of pasta containing 50 g of digestible carbohydrates are shown in Figure 2.

At 15 min after the consumption of the control pasta (CON) and the fortified pasta (LF0, LF20, LFM20), the glucose concentration was significantly (*p* ≤ 0.05) lower than the value achieved after the consumption of the glucose standard solution. Moreover, during the study, neither the fortified pasta nor the control pasta reached their baseline glucose value. The maximum blood glucose concentration (GC) was reached 30 min after consumption of the standard solution and all tested pasta samples. The slight increase in the GC at 120 min after the consumption of the samples LF0 and CON could be related to the higher protein and fat content in these products (compared to the glucose standard solution, which was devoid of these macronutrients). The presence of protein and fat delays the absorption of glucose into the bloodstream due to the longer digestion and slower discharge of food from the stomach into the duodenum [1]. If the time of postprandial glucose testing had been extended to 180 min, the glucose level would also have slightly increased after consumption of the LF20 and LFM20 pasta (with the highest protein and fat content).

The determined postprandial glucose curves allowed for the determination of the Incremental Area Under Curve (IAUC) and the calculation of GI and GL for the pasta samples. The values of the GI and the GL of the studied pasta variants are presented in Table 5. The GI is defined as low if it is less than 55% [3]. The GI for the control samples (durum wheat pasta) was 48%, which was similar to the results reported by other authors [6,45]. Many authors emphasize that pasta is a carbohydrate product with a more beneficial effect on glycemia than bread. The meta-analysis carried out by Huang et al. [46] shows that pasta induces a statistically significant (*p* ≤ 0.05) milder glycemic response than wheat bread and potatoes. Brennan and Tudorica [47] emphasize that the compact structure of pasta with starch granules trapped within the protein matrix is responsible for its relatively low GI. The strong gluten network makes it difficult for amylolytic enzymes to access the starch. In addition, the coarser granulation of the raw material contributes to a higher content of resistant starch (RS1) [1]. Starch enclosed in undamaged cell structures, surrounded by a cellulose skeleton of cell walls, is not subject to enzymatic hydrolysis. Sissons [2] emphasizes that the ratio of amylose and amylopectin in starch is of great importance for the GI of pasta. The authors proved that the increase in the proportion of amylose in durum wheat causes a 35-fold increase in the RS content in the raw material and is reflected in the lower GI and GL values of pasta.

Numerous studies show that enrichment of pasta with high-fiber and high-protein raw materials has a positive effect on postprandial glycemia. However, the type and amount of the additive seem to be essential. The 7.5% addition of β-glucans and the 5% addition of vital gluten (LF0) used in our research caused only a slight decrease (*p* ≤ 0.05) in the pasta GI (Table 5). The enrichment of the pasta with 20% of lupine flour resulted in a significant decrease in the pasta GI. The LF20 sample was characterized by a 20% lower GI compared to the CON variant. The lowest value of the GI (33.75%) was recorded for the LFM 20 sample containing 20% of β-glucans, 20% of lupine flour, and 20% of millet flour. It should be emphasized that the LFM20 sample had the highest concentration of TDF and SDF. Many researchers [47,48] have shown that both the type and level of dietary fiber addition have an impact on the GI of pasta. A considerable role in the GI of products is assigned to the SDF fraction, which includes β-glucans. The positive effect of β-glucans on postprandial glycemia is associated with an increase in the viscosity of the chyme and, as a result, a reduction in the dynamics of digestion and absorption of carbohydrates. As reported by Brennan and Tudorica, the addition of xanthan gum and guar gum has a positive effect on postprandial glycemia [47].

In addition to fiber, the protein content also has a reducing effect on postprandial glycemia. In vivo studies conducted by Chan et al. [49] showed that the replacement of semolina with 25% of flour, isolate, or concentrate of bean proteins had a positive effect on postprandial glycemia and the feeling of satiety. The addition of legume seeds, which are a rich source of protein, reorganizes the starch structure. In a study conducted by Gangola et al. [50], the enrichment of semolina pasta with flour and faba bean protein isolate and concentrate reduced the digestibility of starch in vitro. The authors associate this tendency with a significant increase in the content of protein and dietary fiber, which is rich in resistant starch (RS2). Similar results were obtained by Cutillo et al. [44] by enriching spaghetti with lupine protein isolate. Turco et al. [51] emphasize that the presence of polyphenols, including flavonoids, is involved in the low GI of pasta made from legumes (pea flour, red lentil flour, grass pea flour, chickpea flour) (GI = 20–23.3%), as these compounds may inhibit α-amylase and α-glucosidase, inhibit glucose absorption in the intestine, stimulate insulin secretion, and reduce hepatic glucose output.

In addition to the GI of the product, an important parameter is its glycemic load (GL), which considers the content of digestible carbohydrates present in an average portion of the product. All the tested pasta samples differed significantly in the GL. The standard portion (50 g) of the control pasta (CON) and the pasta enriched only with the addition of β-glucans and vital gluten (LF0) had a GL value in the range of 11–19% (mean GL). The same standard portions of pasta enriched with 20% lupine flour (samples LF20 and LFM20) exhibited GL values below 10%; hence, they can be classified as products with a low GL. These types of products can be a valuable part of the diet for patients with diabetes, overweight and obesity, and metabolic syndrome. During a 12-week dietary intervention in patients with type 2 diabetes, Jenkins et al. [11] found that the increased consumption of legumes in combination with whole-grain cereal products decreased the value of glycated hemoglobin, whose values are correlated with diabetes control. The diet also lowered blood pressure, heart rate, and estimated relative risk of coronary artery disease.

## 4. Conclusions

The present results have proved that the supplementation with lupine flour with simultaneous addition of vital gluten and β-glucan yields a functional product with high cooking and organoleptic properties. The substitution of durum semolina with lupine flour increased the content of protein, fat, ash, and total fiber, as well as the soluble fiber fraction, at the fortification level of 20–25%. The enrichment of pasta significantly reduced the content of digestible carbohydrates. The lupine flour addition up to 20% caused only slight changes in the physical properties and cooking quality of pasta. The use of lupine flour at the level 20% significantly decreased postprandial glycemia and reduced the GI and GL values of the pasta. Pasta enriched with 20% of legumes with a simultaneous 20% addition of β-glucans and millet flour had 30% lower GI and a 58% lower GL values than the control sample. The study shows that the 20% addition of lupine flour allows for a functional product with good-quality properties to be obtained.

## Figures and Tables

**Figure 1 foods-11-03216-f001:**
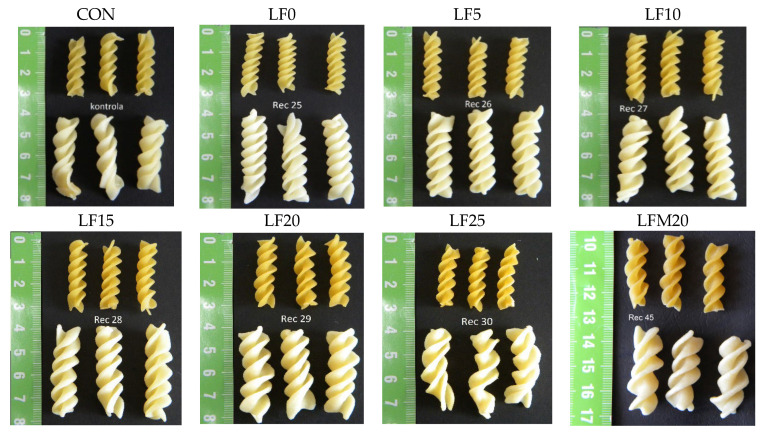
Uncooked (above) and cooked (below) pasta samples; Explanation: CON, control sample; LF, lupin flour; LFM, lupin flour and millet.

**Figure 2 foods-11-03216-f002:**
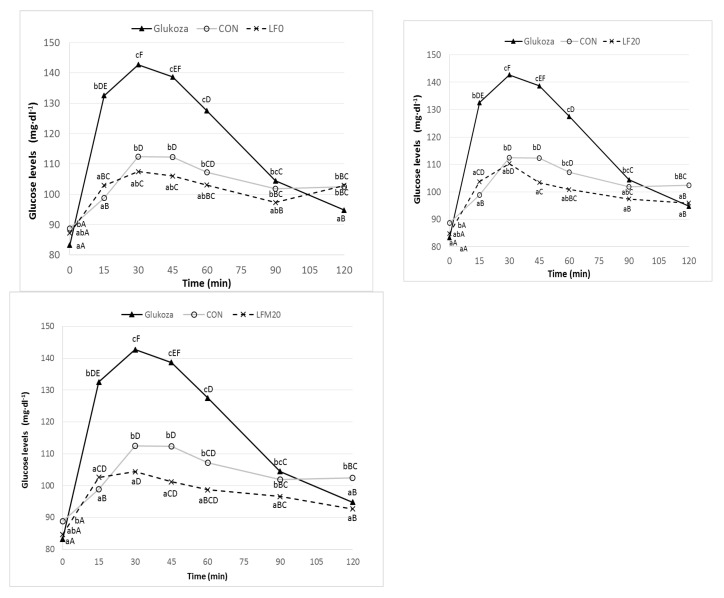
Postprandial glycemic curves after ingestion of standard glucose and pasta samples (CON, LF0, LF20 and LFM20). Explanation: CON, control sample; LF, lupin flour; LFM, lupin flour and millet flour; Data are presented as mean (*n* = 10). Data values with different capital letters indicate statistically significant differences between glucose levels in time after ingestion; data values with different lowercase indicate statistically significant differences between glucose levels between glucose standard and pasta samples (Tukey test, *p* ≤ 0.05).

**Table 1 foods-11-03216-t001:** Model of the experiment.

Pasta Samples	Raw Materials (%)					Moisture (%)	Process Parameters	
							Pressure (MPa)	Barrel Temperature (°C)
	Semolina Durum	Lupin Flour	β-Glucans	Vital Gluten	Millet Flour			
CON	100	0	0	0	0	33	8.5	28.9
LF0	87.5	0	7.5	5	0	31	13.5	28.7
LF5	82.5	5	7.5	5	0	31	13.5	29.1
LF10	77.5	10	7.5	5	0	31	13.5	29.8
LF15	72.5	15	7.5	5	0	31	13.5	29.4
LF20	67.5	20	7.5	5	0	31.5	13.5	29.1
LF25	62.5	25	7.5	5	0	31.5	13.5	29.2
LFM20	35	20	20	5	20	32	13.5	28.9

Explanation: CON, control sample; LF, lupin flour; LFM, lupin flour and millet flour.

**Table 2 foods-11-03216-t002:** Chemical composition of raw material and pasta samples.

Samples	Moisture	Protein	Fat	Ash	TDF	IDF	SDF	Digestible Carbohydrates
	(%)	(% d.m.)						
Raw materials								
Semolina durum	9.50 ± 0.04	13.24 ± 0.84	1.10 ± 0.04	0.76 ± 0.01	3.89 ± 0.14	2.07 ± 0.08	1.83 ± 0.05	81.01 ± 1.01
Lupin flour	7.62 ± 0.07	40.97 ± 2.65	8.13 ± 0.28	2.98 ± 0.03	46.64 ± 2.17	35.75 ± 1.79	10.89 ± 0.48	1.28 ± 0.05
Millet flour	11.14 ± 0.06	14.94 ± 0.93	4.24 ± 0.04	1.34 ± 0.01	12.46 ± 0.54	9.46 ± 0.57	1.81 ± 0.03	67.02 ± 0.89
Oat β-glucans	4.19 ± 0.07	8.45 ± 0.06	3.56 ± 0.08	1.99 ± 0.02	33.48 ± 1.29	0.64 ± 0.19	32.85 ± 1.10	52.51 ± 0.93
Vital gluten	7.31 ± 0.05	70.99 ± 4.61	1.48 ± 0.03	0.57 ± 0.01	23.81 ± 0.69	22.69 ± 0.89	1.12 ± 0.20	3.15 ± 0.21
Pasta samples								
CON	8.83 ^a^ ± 0.23	13.57 ^a^ ± 0.17	1.67 ^a^ ± 0.08	1.06 ^ab^ ± 0.03	4.44 ^a^ ± 0.92	2.10 ^a^ ± 0.65	2.33 ^a^ ± 0.26	79.26 ^a^ ± 1.19
LF0	9.39 ^ab^ ± 0.16	16.57 ^b^ ± 0.62	2.46 ^b^ ± 0.11	0.97 ^a^ ± 0.03	9.33 ^ab^ ± 0.33	5.68 ^ab^ ± 0.23	3.64 ^a^ ± 0.09	70.68 ^b^ ± 1.79
LF5	9.36 ^ab^ ± 0.17	17.63 ^bc^ ± 0.56	2.73 ^b^ ± 0.12	1.07 ^ab^ ± 0.06	10.69 ^b^ ± 1.27	6.24 ^ab^ ± 0.02	4.45 ^a^ ± 1.56	67.88 ^bc^ ± 2.89
LF10	8.82 ^a^ ± 0.16	19.03 ^cd^ ± 0.25	3.05 ^bc^ ± 0.14	1.20 ^bc^ ± 0.08	11.64 ^b^ ± 0.82	6.77 ^b^ ± 0.03	4.87 ^a^ ± 0.79	65.09 ^bc^ ± 2.0
LF15	9.55 ^b^ ± 0.17	20.40 ^de^ ± 0.53	3.35 ^c^ ± 0.16	1.30 ^cd^ ± 0.08	12.99 ^b^ ± 1.31	7.00 ^b^ ± 0.58	5.98 ^a^ ± 0.73	61.96 ^c^ ± 2.78
LF20	9.53 ^b^ ± 0.08	22.37 ^f^ ± 0.31	3.35 ^cd^ ± 0.17	1.35 ^cd^ ± 0.02	17.32 ^c^ ± 1.83	8.16 ^b^ ± 0.45	9.16 ^b^ ± 1.38	51.26 ^d^ ± 0.63
LF25	9.52 ^b^ ± 0.17	22.91 ^f^ ± 0.00	3.99 ^de^ ± 0.19	1.46 ^d^ ± 0.02	20.62 ^d^ ± 1.73	8.09 ^b^ ± 2.45	11.59 ^bc^ ± 0.84	51.02 ^d^ ± 2.64
LFM20	9.21 ^ab^ ± 0.04	20.75 ^e^ ± 0.15	4.86 ^f^ ± 0.23	1.46 ^d^ ± 0.05	25.89 ^e^ ± 0.86	14.31 ^c^ ± 1.59	12.53 ^c^ ± 1.36	47.03 ^e^ ± 1.99

Explanation: IDF, insoluble dietary fiber; SDF, soluble dietary fiber; TDF, total dietary fiber; CON, control sample; LF, lupin flour; LFM, lupin flour and millet flour; Data are presented as mean (*n* = 3) ± standard deviation. Data value of each parameter with different superscript letter in the columns are significantly different (Tukey test, *p* ≤ 0.05).

**Table 3 foods-11-03216-t003:** The apparent viscosity of pasta samples [Pas].

Pasta Samples	Heating				Cooling		
	Temp. 65 °C	Temp. 75 °C	Temp. 85 °C	Temp. 95 °C	Temp. 95 °C *	Temp. 50 °C	Temp. 50 °C **
CON	0.006 ^aA^	0.007 ^aA^	0.010 ^aB^	0.011 ^aB^	0.014 ^aC^	0.015 ^aC^	0.013 ^aC^
LF0	0.009 ^abA^	0.011 ^bcAB^	0.012 ^aABC^	0.013 ^abBC^	0.014 ^aC^	0.015 ^aC^	0.014 ^aC^
LF5	0.009 ^abA^	0.009 ^acA^	0.012 ^aB^	0.012 ^abB^	0.014 ^aBC^	0.015 ^aC^	0.013 ^aBC^
LF10	0.00 ^abA^	0.009 ^acA^	0.011 ^aAB^	0.013 ^abAB^	0.014 ^aAB^	0.015 ^aB^	0.016 ^aB^
LF15	0.008 ^abA^	0.010 ^bcA^	0.012 ^aA^	0.012 ^abA^	0.014 ^aA^	0.015 ^aA^	0.014 ^aA^
LF20	0.010 ^bA^	0.010 ^bcA^	0.012 ^aAB^	0.012 ^abAB^	0.013 ^aAB^	0.014 ^aB^	0.014 ^aB^
LF25	0.009 ^abA^	0.011 ^bcAB^	0.011 ^aAB^	0.013 ^abBC^	0.013 ^aBC^	0.015 ^aC^	0.013 ^aBC^
LFM20	0.009 ^bA^	0.013 ^bAB^	0.016 ^bB^	0.016 ^bB^	0.014 ^aAB^	0.022 ^bC^	0.023 ^bC^

Explanation: * measurement after 20 min, ** measurement after 30 min; CON, control sample; LF, lupin flour; LFM, lupin flour and millet flour; Data are presented as mean ± standard deviation. Data value of each parameter with different uppercase superscript letters in the rows are significantly different (Tukey test, *p* ≤ 0.05). Data value of each parameter with different lowercase superscript letters in the columns are significantly different (Tukey test, *p* ≤ 0.05).

**Table 4 foods-11-03216-t004:** Cooking quality of pasta samples.

Pasta Samples	Optimum Cooking Time (min)	Cooking Loss (% d.m.)	Cooking Weight Increase	Cooking Volume Increase
CON	9 ^a^	4.52 ^abc^ ± 0.42	2.54 ^ab^	3.16 ^a^
LF0	9 ^a^	4.32 ^ab^ ± 0.14	2.46 ^a^	2.68 ^a^
LF5	9 ^a^	4.25 ^a^ ± 0.23	2.51 ^ab^	2.89 ^a^
LF10	9 ^a^	5.25 ^abc^ ± 0.19	2.50 ^ab^	2.95 ^a^
LF15	10 ^ab^	5.71 ^bcd^ ± 0.19	2.54 ^ab^	2.97 ^a^
LF20	11.5 ^cb^	5.85 ^cd^ ± 0.08	2.61 ^bc^	3.10 ^a^
LF25	12 ^c^	7.10 ^de^ ± 0.28	2.70 ^c^	3.13 ^a^
LFM20	12 ^c^	7.43 ^e^ ± 0.30	2.58 ^abc^	2.75 ^a^

Explanation: % d.m-% of dry matter; CON, control sample; LF, lupin flour; LFM, Lupin flour and millet flour. Data are presented as mean ± standard deviation. Data value of each parameter with different superscript letter in the columns are significantly different (Tukey test, *p* ≤ 0.05).

**Table 5 foods-11-03216-t005:** Glycemic index (GI) and glycemic load (GL) of selected pasta samples.

Pasta Samples	GI (%)	% GI Decrease from Control Pasta	GL (%)	% GL Decrease from Control Pasta
CON	48 ^c^	-	17.3 ^d^	-
LF0	46.52 ^c^	3.08	14.9 ^c^	13.87
LF20	38.12 ^b^	20.58	9.2 ^b^	46.82
LFM20	33.75 ^a^	29.68	7.2 ^a^	58.38

Explanation: CON, control sample; LF, lupin flour; LFM, lupin flour and millet flour. Data value of each parameter with different superscript letter in the columns are significantly different (Tukey test, *p* ≤ 0.05).

## Data Availability

The data can be found in the corresponding author of this publication.

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
