# Peer review of "The Effect of the Addition of Low-Alkaloid Lupine Flour on the Glycemic Index In Vivo and the Physicochemical Properties and Cooking Quality of Durum Wheat Pasta"

_foods, 2022, doi:10.3390/foods11203216_

Round 1

Reviewer 1 Report

This work represents a welcome addition to the literature especially as the expensive and time consuming GI test was performed which is rarely done when researchers add ingredients to pasta.

Its is not apparent how the quality compares between control and LF5-25 in discussion & conclusion. Make it clear to the reader that physical pasta CT CL CWI CVI were not that different to CON and acceptable? while providing lower GI. However, authors should comment further on appearance of the cooked pasta in Fig 1 and explain further on the comment "however the appearance of the cooked product (Figure 1) proves that such addition is too high." Is this going to be a major limitation? Reader is left wondering about the best sample to make is it LF25 or LFM20? What patent was applied to which of these samples made (or all of them)?

See annotated comments and corrections in marked up pdf

Author Response

ANSWERS TO THE REVIEWER 1 COMMENTS

We would like to thank the Reviewer for valuable comments which have helped us to improve quality of our work. The answers are listed in the PDF file „foods-1907828-review 1” below the Reviewer’s specific comments. All changes have been updated in the manuscript.

Reviewer 2 Report

General comments:

In this paper, the effect of low-alkaloid lupine flour on the GI values and physical properties of pasta was evaluated, and chemical composition and cooking quality analysis of pasta samples were also made. It’s of high practical value and sufficient data were provided. Adding different contents of lupine flour, quantitative vital gluten, and oat β-glucans would be interesting research, but you have already got Low GI and GI pasta, why chose millet flour, and the recipe is totally new for LFM20? We can associate that this lupine material can reduce the GI of pasta, but further discussion is needed since this paper explains the GI and quality difference from the perspective of different component content of the material, but there may be interactions between different components, which the author has not discussed.

Other comments

1. During pasta preparation, the moisture content of the recipe was set as 33% for CON, 31%-31.5% for LF samples, and 32% for LFM sample. How did the author regulate water? On what basis since water absorption of raw materials varies?

2. In the chemical analysis part, all the changes in components from raw materials to pasta products should be due to the extrusion process. I am not sure whether so many sentences are needed to explain the changes in chemical components, but the impact of the extrusion process is ignored in the discussion of part 3.2. Otherwise, we could calculate the chemical content of samples according to the recipe.

3. The cooking time should be optimum cooking time OCT.

4. What is the taste and flavor fed back by testers during the sensory evaluation?

5. Why just have only one paste sample for millet flour?

Author Response

ANSWERS TO THE REVIEWER 2 COMMENTS

We would like to thank the Reviewer for valuable comments which have helped us to improve quality of our work. The answers to the Reviewer’s specific comments are listed below:

General comments:

In this paper, the effect of low-alkaloid lupine flour on the GI values and physical properties of pasta was evaluated, and chemical composition and cooking quality analysis of pasta samples were also made. It’s of high practical value and sufficient data were provided. Adding different contents of lupine flour, quantitative vital gluten, and oat β-glucans would be interesting research, but you have already got Low GI and GI pasta, why chose millet flour, and the recipe is totally new for LFM20? We can associate that this lupine material can reduce the GI of pasta, but further discussion is needed since this paper explains the GI and quality difference from the perspective of different component content of the material, but there may be interactions between different components, which the author has not discussed.

The aim of the study was to obtain pasta with a GI lower than 40 and a GL lower than 10. When developing the recipes, a variable share (0-25%) of lupine flour was used . Research has shown that a higher addition of legumes flour will significantly deteriorate the quality of pasta. In order to have a greater guarantee that the product will have a sufficiently low GI (<40), the addition of beta-glucans was increased from 7.5 to 20 and the semolina was replaced with a 20% addition of millet flour. Literature data shows that the fat present in the millet flour complexes with amylose. Complexed starch referred to as resistant starch (RS5) can further lower the GI of the product.

Other comments:

  1. During pasta preparation, the moisture content of the recipe was set as 33% for CON, 31%-31.5% for LF samples, and 32% for LFM sample. How did the author regulate water? On what basis since water absorption of raw materials varies?

For the trials with a variable proportion of lupin flour, the addition of water was used to enable the same pressure of pasta dough extrusion. The moisture content of the dough for the control sample was determined on the pilot studies and allowed the highest cooking and sensory  quality of the pasta.

  1. In the chemical analysis part, all the changes in components from raw materials to pasta products should be due to the extrusion process. I am not sure whether so many sentences are needed to explain the changes in chemical components, but the impact of the extrusion process is ignored in the discussion of part 3.2. Otherwise, we could calculate the chemical content of samples according to the recipe.

In the case of the extrusion process (even low-temperature), in opinion of the authors, it is not correct to calculate the content of chemical components based on the chemical composition of the raw materials and their proportion in the recipe.  During the process, complexation of fat with amylose occurs, among other things. The free fat content decreases and the fibre content may increase due to the formation of indigestible amylose-fat complexes described as resistant starch RS5. 

  1. The cooking time should be optimum cooking time OCT.

Change has been made in the manuscript accordingly.

  1. What is the taste and flavor fed back by testers during the sensory evaluation?

Participants used a 0-5 scale. 5 was indicated that the selected features are specific for used raw materials and the most acceptable. 0 meant that smell or taste is nonspecific, with additional aftertastes like sour or musty. There were no significant differences in odor between pasta with the addition of lupine flour or millet flour and the control sample, except for the sample with 25% addition of lupine flour, which was assessed significantly (p ≤ 0.05) lower. Noticeable differences in taste began to appear with 15% enrichment with lupine flour. The taste of the control was rated at an average of “5”, LF20 to “4.2”, while LF25 and LFM20 to “3.8”.

  1. Why just have only one paste sample for millet flour?

Pasta samples were carried out as part of the project (cooperation with the industry). We have pre-tested several samples. For further research and finally for publication, the most representative sample with millet flour was selected, the implementation of which was actually possible in industrial production.
